# Awareness, treatment, and control of hypertension in adults aged 45 years and over and their spouses in India: A nationally representative cross-sectional study

Sanjay K. Mohanty [1]*, Sarang P. Pedgaonkar [2], Ashish Kumar Upadhyay [3], Fabrice Kämpfen [4], Prashant Shekhar [3], Radhe Shyam Mishra [3], Jürgen Maurer [5], Owen O'Donnell [6]

1 Department of Fertility Studies, International Institute for Population Sciences, Mumbai, India, 2 Department of Population Policies and Programmes, International Institute for Population Sciences, Mumbai, India, 3 International Institute for Population Science, Mumbai, India, 4 Population Studies Center, University of Pennsylvania, Philadelphia, Pennsylvania, United States of America, 5 Institute of Health Economics and Management, Department of Economics, University of Lausanne, Switzerland, 6 Erasmus School of Economics & Erasmus School of Health Policy & Management, Erasmus University Rotterdam, Rotterdam, the Netherlands

* sanjayiips@yahoo.co.in

**Data Availability Statement:** Data are publicly available and can be accessed by registering at https://iipsindia.ac.in/sites/default/files/LASI_

## Abstract

### Background

Lack of nationwide evidence on awareness, treatment, and control (ATC) of hypertension among older adults in India impeded targeted management of this condition. We aimed to estimate rates of hypertension ATC in the older population and to assess differences in these rates across sociodemographic groups and states in India.

### Methods and findings

We used a nationally representative survey of individuals aged 45 years and over and their spouses in all Indian states (except one) in 2017 to 2018. We identified hypertension by blood pressure (BP) measurement ≥140/90 mm Hg or self-reported diagnosis if also taking medication or observing salt/diet restriction to control BP. We distinguished those who (i) reported diagnosis ("aware"); (ii) reported taking medication or being under salt/diet restriction to control BP ("treated"); and (iii) had measured systolic BP <140 and diastolic BP <90 ("controlled"). We estimated age–sex adjusted hypertension prevalence and rates of ATC by consumption quintile, education, age, sex, urban–rural, caste, religion, marital status, living arrangement, employment status, health insurance, and state. We used concentration indices to measure socioeconomic inequalities and multivariable logistic regression to estimate fully adjusted differences in these outcomes. Study limitations included reliance on BP measurement on a single occasion, missing measurements of BP for some participants, and lack of data on nonadherence to medication.

The 64,427 participants in the analysis sample had a median age of 57 years: 58% were female, and 70% were rural dwellers. We estimated hypertension prevalence to be 41.9%

DataRequestForm_0.pdf. Data will be made available to the researchers meeting the criteria for access to confidential data. We are enclosing the do files used for tabulation and graph as additional file.

**Funding:** This was supported by funding from the Swiss National Science Foundation (https://www.snf.ch) and the Swiss Agency for Development and Cooperation (https://www.eda.admin.ch/deza/en/home.html) through the Swiss Programme for Research on Global Issues for Development (http://www.r4d.ch) grant 400640_160374: "Inclusive social protection for chronic health problems" (https://r4d-ncd.org) (JM). The content of the Article is solely the responsibility of the authors and does not necessarily represent the views of the Swiss National Science Foundation or the Swiss Agency for Development & Cooperation. The funders had no role in study design, data collection and analysis, decision to publish, or preparation of the manuscript.

**Competing interests:** The authors have declared that no competing interests exist.

**Abbreviations:** ATC, awareness treatment, and control; BP, blood pressure; CI, confidence interval; COVID-19, Coronavirus Disease 2019; CVD, cardiovascular disease; DALY, disability-adjusted life year; IHCI, India Hypertension Control Initiative; IIPS, International Institute for Population Sciences; IQR, interquartile range; LASI, Longitudinal Ageing Study in India; MPCE, monthly per capita consumption expenditure; NCD, noncommunicable disease; NDSP, net state domestic product; pp, percentage points; SDP, state domestic product; STROBE, Strengthening the Reporting of Observational Studies in Epidemiology; WHO, World Health Organization.

(95% CI 41.0 to 42.9). Among those with hypertension, we estimated that 54.4% (95% CI 53.1 to 55.7), 50.8% (95% CI 49.5 to 52.0), and 28.8% (95% CI 27.4 to 30.1) were aware, treated, and controlled, respectively. Across states, adjusted rates of ATC ranged from 27.5% (95% CI 22.2 to 32.8) to 75.9% (95% CI 70.8 to 81.1), from 23.8% (95% CI 17.6 to 30.1) to 74.9% (95% CI 69.8 to 79.9), and from 4.6% (95% CI 1.1 to 8.1) to 41.9% (95% CI 36.8 to 46.9), respectively. Age–sex adjusted rates were lower ($p < 0.001$) in poorer, less educated, and socially disadvantaged groups, as well as for males, rural residents, and the employed. Among individuals with hypertension, the richest fifth were 8.5 percentage points (pp) (95% CI 5.3 to 11.7; $p < 0.001$), 8.9 pp (95% CI 5.7 to 12.0; $p < 0.001$), and 7.1 pp (95% CI 4.2 to 10.1; $p < 0.001$) more likely to be aware, treated, and controlled, respectively, than the poorest fifth.

## Conclusions

Hypertension prevalence was high, and ATC of the condition were low among older adults in India. Inequalities in these indicators pointed to opportunities to target hypertension management more effectively and equitably on socially disadvantaged groups.

## Author summary

### Why was this study done?

- We found only one study that reported estimated rates of awareness, treatment, and control (ATC) of hypertension in India using a nationally representative sample covering all states, but that study was restricted to adults aged 15 to 49 years.

- Another study estimated rates of hypertension ATC among older adults, but that study covered only 6 states.

- This study aimed to provide nationally representative estimates of hypertension ATC in the older population of India and to describe differences in these indicators of hypertension management across sociodemographic groups and states.

### What did the researchers do and find?

- We used a nationally representative sample of adults aged 45 years and over and their spouses covering all states (except one) of India in 2017 to 2018.

- We used measured blood pressure (BP) and self-reported diagnosis and treatment for high BP to estimate hypertension prevalence and the percentages of those with hypertension who were aware of their condition, treated for it, and had achieved BP control.

- We found that a slight majority of those with hypertension were aware of their condition, around half were being treated, and less than a third had controlled their BP. While these rates indicated substantial gaps in hypertension management among the older population of India, they were higher than estimates previously obtained from samples restricted to, or including, younger people.

- We found substantial variation in the indicators of hypertension management across states. Older Indians who were poorer, less educated, socially disadvantaged, male, rural, and working were less likely to be aware, treated, and to have achieved BP control.

## What do these findings mean?

- Hypertension prevalence is high in India, particularly in the older population. In this critical population group, low rates of ATC point to deficiencies in diagnosis and management of the condition and in the prevention of cardiovascular diseases (CVDs).

- Effectively addressing these deficiencies requires subtle targeting of interventions that balances attention to prevalence, which is higher in the high-income states and socio-economically advantaged groups, with attention to gaps in ATC, which are greater in the low- or middle-income states and disadvantaged groups.

## Introduction

Hypertension is a major risk factor for cardiovascular diseases (CVDs) [1,2] that accounted for 44% of the 42 million deaths related to noncommunicable diseases (NCDs) globally in 2019 [3]. Better management of hypertension is critical to accelerating progress toward the Sustainable Development Goal target of a one-third reduction in premature NCD-related mortality by 2030 [4]. To maximize the impact on global health, improvements in care for hypertension need to occur where prevalence is increasing most rapidly, such as in South Asia [5], and where populations are largest, such as in India, which accounts for 18% of global population [6]. These improvements generally need to be targeted on middle-aged and older adults who are at greatest risk.

India is undergoing demographic and epidemiological transitions, as well as economic growth and urbanization, which make targeted, effective hypertension management a population health imperative. The older population aged 45 years and over has been growing twice as fast as the overall population [7], and the NCD share of disability-adjusted life years (DALYs) reached 58% in 2019 [3]. In 2012 to 2014, one quarter of the adult population was estimated to have hypertension, and the prevalence was almost twice as high among older adults aged 65+ [8]. Prevalence varied widely across Indian states and was found to be higher in wealthier groups [8]. The median age of onset of hypertension is estimated to have declined from 61 years in 2004 to 55 years by 2018 [9]. Hypertension is the main risk factor of CVD in the country [10] and accounted for 35.5% of DALYs in 2019, compared with 26.8% globally [3].

The Government of India was quick to adopt the World Health Organization (WHO) Global Action Plan for the Prevention and Control of NCDs [11], targeted a 25% reduction in hypertension prevalence between 2013 and 2025 [12], and committed to population-based screening and management for hypertension and other NCD risk factors [12]. Monitoring the success of such programs, and targeting them effectively, requires evidence on rates of diagnosis, treatment, and blood pressure (BP) control among those with hypertension. However, to our knowledge, there were no estimates of hypertension awareness, treatment, and control (ATC) obtained from a nationally representative sample of the middle-aged and older population throughout the whole of India. Estimates from a nationally representative sample of the

population aged 18 to 49 [13] and from samples that included older adults but did not have nationwide coverage [14–20] did not permit, without imposing strong assumptions, evaluation of hypertension management across the country in the age group most prone to the condition and its related risks. Lack of evidence on variation in hypertension ATC among middle-aged and older (hereafter, "older") adults across states and sociodemographic groups limited assessment of equity and effectiveness of hypertension care and constrained ability to target interventions on the subpopulations of older adults that have the greatest potential to gain.

This study aimed to estimate levels of hypertension ATC in the older population of India and to assess inequalities in these indicators of hypertension management across states and sociodemographic groups.

## Methods

### Study design and participants

We used data from the Longitudinal Ageing Study in India (LASI) conducted from April 2017 to December 2018. The study used stratified, multistage probability cluster random sampling to select 42,949 households (S1 Text) [21]. In these households, all individuals aged 45 years and over and their spouses were targeted for interview. The survey was conducted in all states and union territories (hereafter, "states"), but data from Sikkim were not available for this study. The sample was representative of the non-institutionalized population aged 45 years and over (and spouses) at the state level, as well as nationally. The response rate was 95.8% at the household level and 87.3% at the individual level [21]. LASI obtained ethical approval from the Health Ministry's Screening Committee (Government of India) and the Institutional Review Boards at the International Institute for Population Sciences (IIPS) and its collaborating institutions. Written informed consent was obtained from all study participants. We followed the Strengthening the Reporting of Observational Studies in Epidemiology (STROBE) guideline in reporting the study (S1 Checklist). We did not prespecify the statistical analysis.

### Measures

BP was measured using an Omron HEM 7121 automatic digital BP monitor manufactured by Omron Healthcare Vietnam Co. Ltd, Vietnam, on a single home visit after completion of the survey questionnaire and before measurement of any other biomarker. Participants were requested to avoid exercise, food, alcohol, and smoking 30 minutes prior to measurement. Three measurements were taken, with a 1-minute gap between each. We used the average of the last 2 readings. We classified a participant as having hypertension if (a) they had systolic BP ≥140 mm Hg or diastolic BP ≥90 mm Hg; or (b) they reported ever having been told they had hypertension/high BP, and they reported currently taking medication or being under salt/diet restriction to control BP (S2 Text). We classified participants with hypertension as (a) "aware" if they reported having been diagnosed with hypertension; (b) "treated" if they reported currently taking medication or being under salt/diet restriction to control BP; and (c) "controlled" if they had systolic BP <140 mm Hg and diastolic BP <90 mm Hg using the survey measurement of BP (S2 Text). The BP thresholds used to define hypertension and hypertension control are those specified in the training module of the Government of India National Programme for Prevention and Control of Cancer, Diabetes, Cardiovascular Diseases, and Stroke [22].

We used household monthly per capita consumption expenditure (MPCE) as the primary indicator of socioeconomic status. This is the measure used to determine a household's poverty status in India [23]. We deviated from the standard measure by excluding expenditure on healthcare and medicines to avoid giving the false impression that households with such

expenditures enjoyed a higher standard of living. We constructed MPCE from detailed reports of items of expenditure and consumption, including goods produced by the household (S3 Text).

## Statistical analysis

We used the full sample of individuals aged 45 years and over and their spouses—who could be younger than 45—to estimate hypertension prevalence. We used a subsample of these participants who were identified as having hypertension to estimate rates of ATC (S2 Text, S4 and S5 Tables). We estimated prevalence and these rates nationally, by state, by MPCE quintile group (S3 Text), and by other sociodemographic characteristics (years of schooling, age group, sex, urban–rural, caste, religion, marital status, living arrangement, employment status, and health insurance status). Years of schooling were categorized into 4 groups corresponding to education levels: no school, incomplete primary school (<5 years), primary high school (5 to 9 years), and secondary school and above (≥10 years). We adjusted estimates for age and sex using the age–sex composition of the nationally representative full sample as the reference (S4 Text).

At the state level, we used a scatter plot and a linear regression line to examine the association between hypertension prevalence and economic development measured by net state domestic product (NSDP) per capita [24]. We used the same analysis to examine associations between rates of ATC and NSDP per capita. We have excluded 3 union territories (Dadra and Nagar Haveli, Daman and Diu, and Lakshadweep) from these analyses due to unavailability of NSDP per capita for the reference period 2017 to 2018.

We used a concentration index—the scaled covariance between an outcome and rank in the MPCE distribution [25]—to quantify the degree of socioeconomic inequality along the full distribution of MPCE. This approach avoided loss of information from grouping participants into MPCE groups such as quintiles and making comparisons only between extremes, e.g., richest fifth versus poorest fifth. A positive (negative) concentration index would indicate, for example, higher prevalence among richer (poorer) individuals. We adjusted the concentration indices for age and sex (S4 Text). We used multivariable logistic regressions to estimate fully adjusted marginal effects of MPCE quintile groups, sociodemographic characteristics, and state indicators on the probability of hypertension, and on the probabilities of ATC among those with hypertension. Each marginal effect was averaged over the sample used in the respective regression.

In all analyses, we applied sampling weights and took account of stratification and cluster sampling in the estimation of confidence intervals (CIs). We included in the analysis sample all participants with full item response on BP measurement, reported diagnosis and treatment for hypertension/high BP, MPCE, and all reported covariates. All the analyses were done using Stata 15.0. The Stata do file is shown in S1 Code.

## Results

Fig 1 shows our selection of the analysis sample. Of the 72,250 participants interviewed, BP could not be measured for 6,499 (9.0%) because participants did not give consent, they were interviewed by proxy, or they had irritations on both arms that prevented fitting the cuff of the BP monitor. Participants for whom BP could not be measured were richer, better educated, and older, and they were more likely to be urban, Muslim, not working, and belong to a privileged caste (S1 Table). A further 1,324 participants (2.0%) did not report all information on hypertension diagnosis and treatment, MPCE, and sociodemographic characteristics (S2 Table), leaving an analysis sample of 64,427 participants with full item response that was used

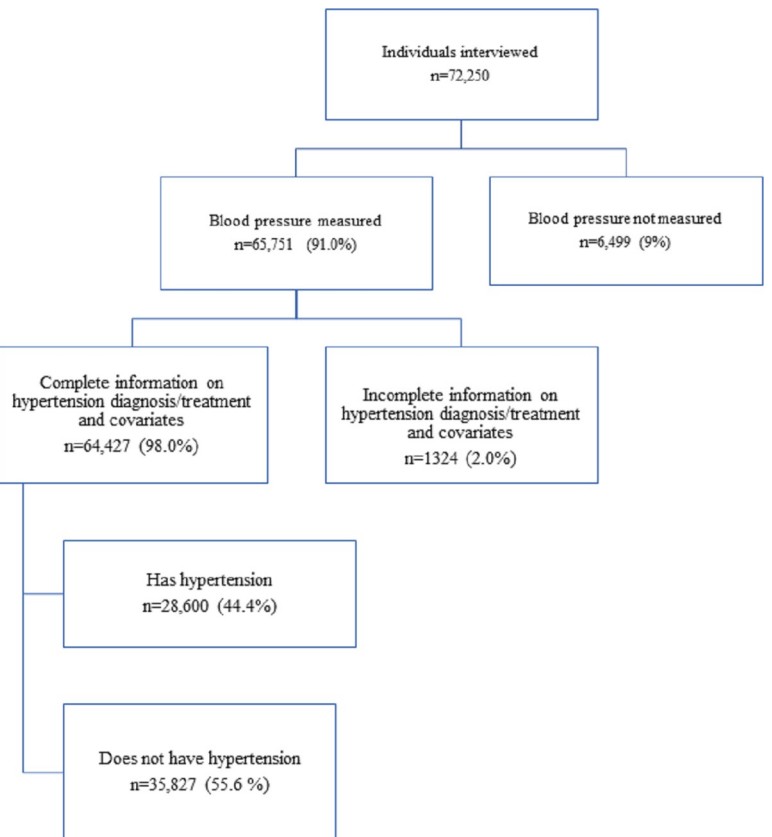

**Fig 1. Flowchart of participant selection.** Percentage are unweighted.

to estimate hypertension prevalence. In this sample, 28,600 participants were identified as having hypertension and were used to estimate rates of hypertension ATC. Estimates of hypertension prevalence and of rates of ATC obtained from the full item response analysis sample used were very close to estimates obtained from a sample that did not exclude those with missing information on MPCE or any sociodemographic characteristic (S3 Table).

Table 1 shows characteristics of the full analysis sample and estimates of age–sex adjusted hypertension prevalence. The median age of the sample was 57 years (interquartile range [IQR]: 49 to 65). A majority was female (58%). Educational attainment was low: half had no formal schooling. The sample was predominantly rural (70%) and married (76%). Less than half of the participants (47%) were working. Only 21% had health insurance.

We estimated hypertension prevalence to be 41.9% (95% CI 41.0 to 42.9) among adults aged 45 years and over and their spouses in India. Prevalence increased strongly with age and was higher for females (43.7%: 95% CI 42.8 to 44.6) than for males (39.6%: 95% CI 38.4 to 40.9) (adjusted for age). Adjusted for age and sex, estimated prevalence increased monotonically in moving from the poorest fifth (35.3%: 95% CI 33.6 to 37.0) to the richest fifth (50.7%: 95% CI 48.2 to 53.2). A positive concentration index of 0.061 (95% CI 0.048 to 0.073) confirmed that richer individuals with higher MPCE were more likely to have hypertension (Table 2). Prevalence was estimated to increase from 37.0% (95% CI 36.0 to 38.0) among the least educated to 51.2% (95% CI 49.1 to 53.2) among the most educated. Prevalence was higher in urban areas (51.8%: 95% CI 50.3 to 53.4) than in rural areas (37.8%: 95% CI 36.9 to 38.7). It was also higher among those in the privileged "other castes" and those not working.

**Table 1. Participant characteristics and adjusted hypertension prevalence, adults aged 45 years and over and their spouses.**

| | Participants | | Adjusted hypertension prevalence | F statistic |
|---|---|---|---|---|
| | *n* | (%) | % (95% CI) | (*p*-value) |
| Overall | 64,427 | 100 | 41.9 (41.0–42.9) | |
| **MPCE quintile group** | | | | |
| Poorest | 11,237 | (20.0) | 35.3 (33.6–37.0) | 25.63 (<0.001) |
| Poorer | 11,514 | (20.0) | 39.5 (38.0–41.0) | |
| Middle | 12,348 | (20.0) | 41.0 (39.4–42.5) | |
| Richer | 14,050 | (20.0) | 43.4 (41.8–45.0) | |
| Richest | 15,278 | (20.0) | 50.7 (48.2–53.2) | |
| **Education** | | | | |
| No schooling | 29,676 | (49.9) | 37.0 (36.0–38.0) | 72.45 (<0.001) |
| <5 years | 7,309 | (11.1) | 43.0 (41.1–44.9) | |
| 5–9 years | 15,240 | (21.3) | 45.6 (44.2–46.9) | |
| ≥10 years | 12,202 | (17.7) | 51.2 (49.1–53.2) | |
| **Age** | | | | |
| <45 years | 6,027 | (8.9) | 22.3 (18.4–26.3) | 164.23(<0.001) |
| 45–54 | 21,573 | (31.7) | 34.5 (33.2–35.7) | |
| 55–64 | 18,077 | (27.8) | 44.2 (42.9–45.6) | |
| 65–74 | 12,994 | (21.8) | 52.4 (50.6–54.3) | |
| ≥75 | 5,756 | (9.8) | 54.5 (52.4–56.6) | |
| **Sex** | | | | |
| Male | 27,154 | (42.1) | 39.6 (38.4–40.9) | 45.25 (0.037) |
| Female | 37,273 | (57.9) | 43.7 (42.8–44.6) | |
| **Location** | | | | |
| Rural | 42,109 | (70.4) | 37.8 (36.9–38.7) | 230.22 (<0.001) |
| Urban | 22,318 | (29.6) | 51.8 (50.3–53.4) | |
| **Caste** | | | | |
| Scheduled caste | 10,894 | (19.7) | 39.0 (37.6–40.4) | 25.48 (<0.001) |
| Scheduled tribe | 11,231 | (8.6) | 36.8 (34.1–39.4) | |
| Other Backward Class | 24,394 | (45.3) | 41.9 (40.3–43.5) | |
| Other | 17,908 | (26.4) | 45.9 (44.6–47.3) | |
| **Religion** | | | | |
| Hindu | 47,255 | (82.4) | 40.8 (39.7–42.0) | 14.66 (<0.001) |
| Muslim | 7,675 | (11.2) | 46.8 (43.6–50.0) | |
| Christian | 6,484 | (2.9) | 43.2 (38.3–48.1) | |
| Other | 3,013 | (3.5) | 51.6 (48.0–55.2) | |
| **Marital status** | | | | |
| Married | 49,632 | (76.2) | 41.0 (39.9–42.1) | 10.77 (<0.001) |
| Widowed | 12,857 | (21.2) | 45.7 (43.8–47.6) | |
| Other | 1,938 | (2.7) | 38.7 (33.4–44.0) | |
| **Living arrangement** | | | | |
| Alone | 2,097 | (3.5) | 44.3 (40.6–47.9) | 5.60 (0.001) |
| With spouse | 9,329 | (15.5) | 40.6 (38.4–42.8) | |
| With children | 39,522 | (59.8) | 41.1 (39.7–42.5) | |
| With others | 13,479 | (21.3) | 44.8 (43.3–46.3) | |
| **Working status** | | | | |

(*Continued*)

**Table 1.** (Continued)

| | Participants | | Adjusted hypertension prevalence | F statistic |
|---|---|---|---|---|
| | *n* | (%) | % (95% CI) | (*p*-value) |
| Working | 29,619 | (47.2) | 38.0 (36.3–39.7) | 26.08 (<0.001) |
| Worked previously | 15,845 | (25.6) | 45.8 (44.5–47.2) | |
| Never worked | 18,963 | (27.3) | 45.0 (43.4–46.6) | |
| **Health insurance** | | | | |
| No | 49,394 | (79.0) | 41.5 (40.6–42.4) | 5.18 (0.023) |
| Yes | 15,033 | (21.0) | 43.7 (41.8–45.6) | |

Notes: Adjusted for age and sex. Prevalence estimates by age group were adjusted for sex composition only. Estimates by sex were adjusted for age composition only. Unadjusted prevalence rates presented in S6 Table. F statistic for test of equal prevalence across groups defined by each characteristic. Other religion includes Sikh, Buddhist/neo-Buddhist, Jain, Jewish, Parsi, and no religion.

MPCE, monthly per capita consumption expenditure.

Table 3 shows adjusted percentages of adults aged 45 years and over and their spouses with hypertension who (a) were aware of their condition; (b) were under treatment for it; and (c) had their BP under control. Overall, we estimated that 54.4% (95% CI 53.1 to 55.7) were aware, 50.8% (95% CI 49.5 to 52.0) were treated, and only 28.8% (95% CI 27.4 to 30.1) had achieved control. There was an 18 percentage points (pp) gap between the richest fifth and the poorest fifth in awareness. The rich–poor gaps were 19 pp and 13 pp for treatment and control, respectively. Concentration indices for ATC were all positive: 0.072 (95% CI 0.056 to 0.089), 0.077 (95% CI 0.061 to 0.093), and 0.055 (95% CI 0.043 to 0.067), respectively (Table 2). This confirmed pro-rich inequalities in hypertension ATC. Gaps between the most and least educated groups were 18 pp, 19 pp, and 13 pp for rates of hypertension ATC, respectively.

Among those with hypertension, males were less likely than females to be aware, treated, and controlled. Rates of ATC were also lower for scheduled tribes, rural dwellers, the employed, those living alone, and those not married or widowed. Younger study participants were less likely to have been diagnosed and treated, but they were not less likely to have achieved BP control. Hindus and Christians were also less likely than those of other religions to be aware and treated. There were no differences in ATC by health insurance status.

Using the full analysis sample that includes participants without hypertension, we estimated that 19.1% (95% CI 18.5 to 19.8) had measured high BP ($\geq$140/90 mm Hg) but had not been diagnosed with hypertension (S7 Table). This prevalence of undiagnosed hypertension was higher among both the poorest and the richest compared with those in the middle. It was higher for the least educated compared to those with more education. The prevalence of

**Table 2. Adjusted concentration indices for hypertension and for ATC among those with hypertension, adults aged 45 years and over and their spouses.**

| | Concentration index (95% CI) | *n* |
|---|---|---|
| Hypertension | 0.061 (0.048–0.073) | 64,427 |
| Awareness | 0.072 (0.056–0.089) | 28,600 |
| Treatment | 0.077 (0.061–0.093) | 28,600 |
| Control | 0.055 (0.043–0.067) | 28,600 |

Notes: A concentration index is a scaled covariance between an outcome and rank in the distribution of MPCE. Adjusted for age and sex. Concentration curves presented in S1 Fig.

ATC, awareness, treatment, and control; MPCE, monthly per capita consumption expenditure.

**Table 3. Adjusted percent aware, treated, and controlled among those with hypertension, adults aged 45 years and over and their spouses.**

| | Awareness | | Treatment | | Control | |
|---|---|---|---|---|---|---|
| ($n$ = 28,600) | % (95% CI) | F statistic ($p$-value) | % (95% CI) | F statistic ($p$-value) | % (95% CI) | F statistic ($p$-value) |
| **Overall** | 54.4 (53.1–55.7) | | 50.8 (49.5–52.0) | | 28.8 (27.4–30.1) | |
| **MPCE quintile group** | | | | | | |
| Poorest | 43.6 (41.0–46.1) | 38.64 (<0.001) | 40.0 (37.6–42.4) | 42.45 (<0.0010) | 22.0 (19.8–24.2) | 16.57 (<0.001) |
| Poorer | 49.5 (47.0–52.0) | | 45.3 (42.7–47.9) | | 24.4 (22.6–26.3) | |
| Middle | 54.9 (52.6–57.3) | | 50.9 (48.5–53.2) | | 28.1 (26.4–29.9) | |
| Richer | 59.0 (56.9–61.1) | | 55.4 (53.3–57.5) | | 32.1 (29.9–34.3) | |
| Richest | 61.7 (59.2–64.1) | | 59.0 (56.7–61.3) | | 34.9 (32.0–37.7) | |
| **Education** | | | | | | |
| No schooling | 47.0 (45.2–48.8) | 57.27 (<0.001) | 43.2 (41.5–45.0) | 62.29 (<0.001) | 24.3 (22.9–25.7) | 16.14 (<0.001) |
| <5 years | 55.2 (52.2–58.2) | | 51.3 (48.3–54.3) | | 28.8 (26.3–31.3) | |
| 5–9 years | 59.5 (57.4–61.7) | | 56.6 (54.4–58.8) | | 31.3 (29.1–33.5) | |
| ≥10 years | 65.4 (62.6–68.3) | | 62.2 (59.5–64.8) | | 37.6 (34.2–40.9) | |
| **Age** | | | | | | |
| <45 years | 40.9 (31.2–50.5) | 11.84 (<0.001) | 37.6 (28.7–46.6) | 9.99 (<0.001) | 24.6 (18.6–30.6) | 0.92 (0.449) |
| 45–54 | 48.2 (45.8–50.7) | | 44.7 (42.3–47.1) | | 26.8 (24.8–28.8) | |
| 55–64 | 55.2 (52.8–57.7) | | 52.0 (49.5–54.5) | | 29.8 (27.5–32.0) | |
| 65–74 | 59.5 (57.0–61.9) | | 55.9 (53.4–58.4) | | 30.3 (27.3–33.3) | |
| ≥75 | 59.2 (56.3–62.1) | | 54.5 (51.5–57.5) | | 28.7 (25.6–31.9) | |
| **Sex** | | | | | | |
| Male | 48.2 (46.3–50.0) | 89.76 (<0.001) | 44.5 (42.7–46.2) | 98.57 (<0.001) | 24.3 (23.1–25.6) | 69.94 (<0.001) |
| Female | 58.8 (57.3–60.4) | | 55.3 (53.7–56.9) | | 31.9 (30.2–33.7) | |
| **Location** | | | | | | |
| Rural | 49.4 (47.8–51.0) | 128.63 (<0.001) | 45.6 (44.0–47.2) | 144.44 (<0.001) | 25.4 (24.1–26.6) | 44.89 (<0.001) |
| Urban | 63.1 (61.2–65.1) | | 59.9 (58.1–61.8) | | 34.7 (32.3–37.1) | |
| **Caste** | | | | | | |
| Scheduled caste | 52.0 (49.5–54.4) | 50.29 (<0.001) | 48.0 (45.4–50.5) | 57.27 (<0.001) | 26.4 (24.2–28.6) | 31.34 (<0.001) |
| Scheduled tribe | 36.0 (32.4–39.5) | | 32.2 (28.9–35.5) | | 16.9 (14.3–19.5) | |
| Other Backward Class | 54.2 (52.4–56.0) | | 50.9 (49.1–52.7) | | 29.6 (27.3–31.8) | |
| Others | 60.8 (59.0–62.7) | | 57.1 (55.2–59.0) | | 31.9 (30.3–33.6) | |
| **Religion** | | | | | | |
| Hindu | 53.2 (51.7–54.6) | 8.95 (<0.001) | 49.5 (48.1–51.0) | 9.26 (<0.001) | 28.4 (26.9–30.0) | 1.98 (0.115) |
| Muslim | 60.2 (57.4–63.0) | | 56.7 (53.9–59.4) | | 31.5 (28.9–34.2) | |
| Christian | 53.6 (48.7–58.5) | | 50.5 (45.7–55.3) | | 26.7 (22.6–30.7) | |
| Others | 60.7 (56.4–65.1) | | 57.1 (52.8–61.3) | | 28.1 (24.8–31.4) | |
| **Marital status** | | | | | | |
| Married | 55.0 (53.3–56.8) | 4.94 (0.007) | 51.6 (49.9–53.3) | 4.79 (0.008) | 29.6 (28.0–31.1) | 6.58 (0.001) |
| Widowed | 53.4 (51.3–55.5) | | 49.4 (47.3–51.5) | | 27.3 (25.3–29.4) | |
| Other | 45.0 (38.6–51.4) | | 42.6 (36.6–48.6) | | 21.4 (16.7–26.1) | |
| **Living arrangement** | | | | | | |
| Alone | 49.7 (45.2–54.3) | 2.94 (0.032) | 46.2 (41.7–50.7) | 3.10 (0.026) | 25.2 (21.2–29.2) | 2.51 (0.057) |
| With spouse | 53.2 (50.4–56.0) | | 49.9 (47.0–52.7) | | 27.9 (25.7–30.1) | |
| With children | 55.6 (53.9–57.4) | | 52.1 (50.4–53.9) | | 30.2 (28.3–32.1) | |
| With others | 53.1 (51.1–55.2) | | 49.3 (47.3–51.2) | | 27.1 (25.0–29.2) | |
| **Working status** | | | | | | |

*(Continued)*

**Table 3.** (Continued)

| | Awareness | | Treatment | | Control | |
|---|---|---|---|---|---|---|
| | % (95% CI) | F statistic (*p*-value) | % (95% CI) | F statistic (*p*-value) | % (95% CI) | F statistic (*p*-value) |
| | | | | | | |
| Working | 44.2 (42.1–46.3) | 87.58 (<0.001) | 40.9 (38.9–42.9) | 83.12 (<0.001) | 23.6 (22.0–25.3) | 30.22 (<0.001) |
| Worked previously | 59.1 (57.0–61.1) | | 54.9 (52.9–56.9) | | 30.1 (28.2–32.0) | |
| Never worked | 63.5 (61.1–65.9) | | 59.9 (57.3–62.5) | | 34.0 (30.8–37.3) | |
| **Health insurance** | | | | | | |
| No | 54.3 (52.9–55.8) | 0.02 (0.876) | 50.6 (49.2–52.1) | 0.20 (0.652) | 28.8 (27.3–30.3) | 0.05 (0.815) |
| Yes | 54.5 (52.3–56.8) | | 51.2 (49.0–53.4) | | 28.6 (26.7–30.4) | |

Notes: Adjusted for age and sex. Estimates by age group were adjusted for sex composition only. Estimates by sex were adjusted by age composition only. Unadjusted estimates presented in S6 Table. F statistic for test of equal rates across groups defined by each characteristic. Number of participants in each sociodemographic category presented in S4 Table.

MPCE, monthly per capita consumption expenditure.

untreated hypertension (BP $\geq$140/90 mm Hg and not on medication/diet) was 20.7% (95% CI 20.0 to 21.3) in the full sample and was not related to MPCE or education (S7 Table). We estimated that 29.9% (95% CI 29.2 to 30.6) of the population had uncontrolled hypertension, and the prevalence was higher for richer and better educated groups (S7 Table).

Fig 2 shows variation across states in adjusted hypertension prevalence and in rates of ATC among those with hypertension. Prevalence exceeded the national average of 42% in 28 of the 35 states and varied from 31.3% (95% CI 29.2 to 33.5) in Uttar Pradesh to 66.0% (95% CI 61.3 to 70.6) in Lakshadweep. Awareness among those with hypertension varied from 27.5% (95% CI 22.2 to 32.8) in Nagaland to 75.9% (95% CI 70.8 to 81.1) in Jammu and Kashmir. Treatment among those with hypertension varied from 23.8% (95% CI 17.6 to 30.1) in Nagaland to 74.9% (95% CI 69.8 to 79.9) in Jammu and Kashmir. In 22 out of 35 states, the estimated percentage of hypertension cases with controlled BP was less than the national average (28.8%). The percentage with controlled BP varied from 4.6% (95% CI 1.1 to 8.1) in Nagaland to 41.9% (95% CI 36.8 to 46.9) in Goa.

S2 Fig presents the adjusted percent treated among those aware of hypertension and adjusted percent controlled among those treated, adults aged 45 years and over, and their spouses in India.

Fig 3 shows that hypertension prevalence was positively associated with NSDP per capita. On average, rates of ATC were also higher in the high-income states. Consequently, states that had higher hypertension prevalence tended to have higher rates of ATC, although the correlation with rates of control was not significant (S10 Table). The positive association with state domestic product (SDP) per capita was sequentially stronger for awareness (R-squared = 0.227), treatment (R-squared = 0.239), and control (R-squared = 0.271). Jammu and Kashmir stands out for having achieved high rates of ATC despite having low SDP per capita, while Arunachal Pradesh, Chhattisgarh, and Nagaland performed even worse than would be expected on the basis of the low level of development in each of these states. Delhi underperformed, particularly on control of hypertension, relative to the linear prediction from its high SDP per capita.

Fig 4 shows adjusted concentration indices for hypertension and for ATC among those with hypertension by state, ordered from the lowest (most prevalent among poor) to the highest (most prevalent among rich) concentration index for hypertension. In all except 3 states, the point estimate of this index is positive, indicating a disproportionate concentration of hypertension among the economically better off, although most of the 95% CIs include zero,

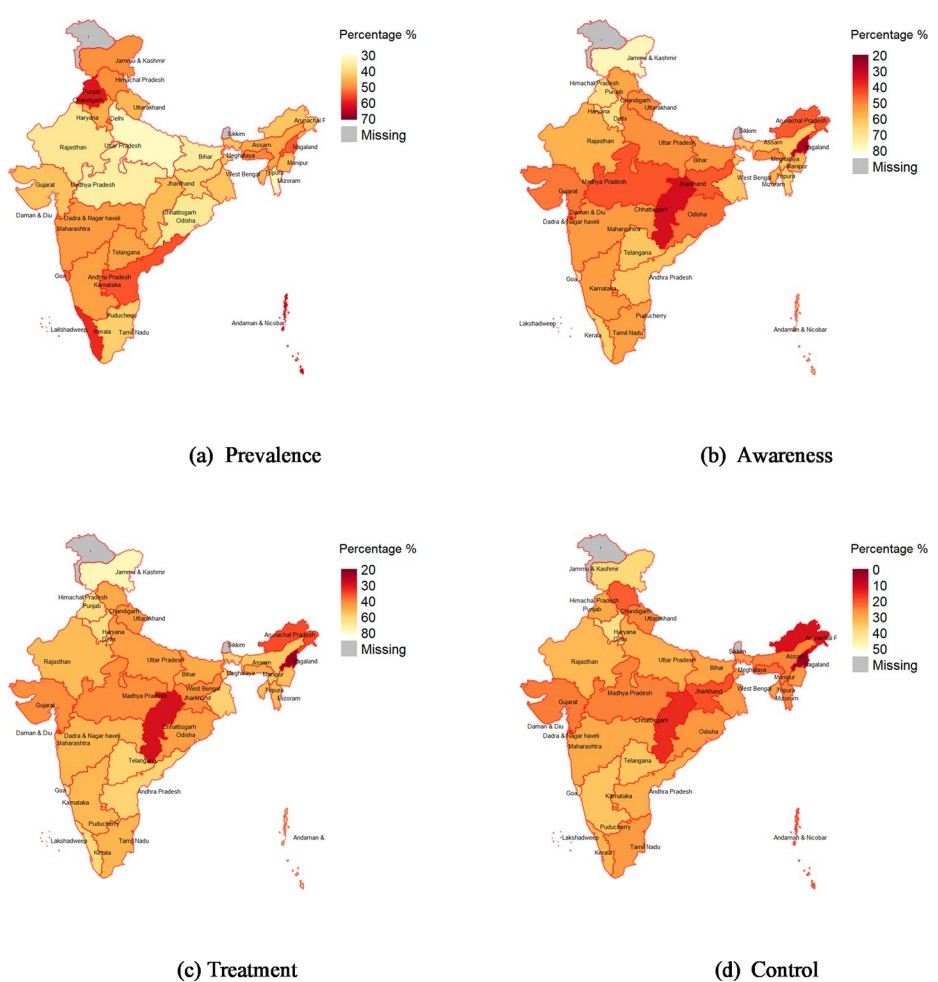

**Fig 2. Adjusted hypertension prevalence and percent aware, treated, and controlled among those with hypertension by state, adults aged 45 years and over and their spouses.** The base map can be found at https://globalsolaratlas.info/download/india. Notes: Adjusted for age and sex. State-specific estimates of age sex adjusted hypertension prevalence and rates of ATC in table form are presented in S8 Table, and unadjusted estimates are presented in S9 Table. ATC, awareness, treatment, and control.

which is consistent with no inequality. With only a few exceptions, the point estimates of the concentration indices for ATC are positive, which indicates that the better off were more likely to have their hypertension diagnosed, treated, and controlled in almost all states. Many states with little or no inequality in the distribution of hypertension had inequalities in ATC that favored the better off. States with greater socioeconomic inequality in prevalence tended to have greater inequality in awareness (Spearman rank correlation ρ = 0.444, *p*-value = 0.008) and treatment (Spearman rank correlation ρ = 0.416, *p*-value = 0.013).

Fig 5 shows average marginal effects from multivariable logistic regressions for each outcome. Controlling for sociodemographic characteristics and state, the prevalence of hypertension was estimated to be 6.1 pp (95% CI 3.0 to 9.2) higher in the richest fifth of the population than in the poorest fifth. In the other MPCE quintile groups, prevalence was estimated to be about 2.3 to 2.5 pp higher than in the poorest fifth. The education gradient in hypertension prevalence remained strong after controlling for other characteristics and state, with a difference of 6.9 pp (95% CI 4.5 to 9.3) between the highest and lowest education groups. Greater prevalence among older adults, women, urban dwellers, and the nonemployed also remained

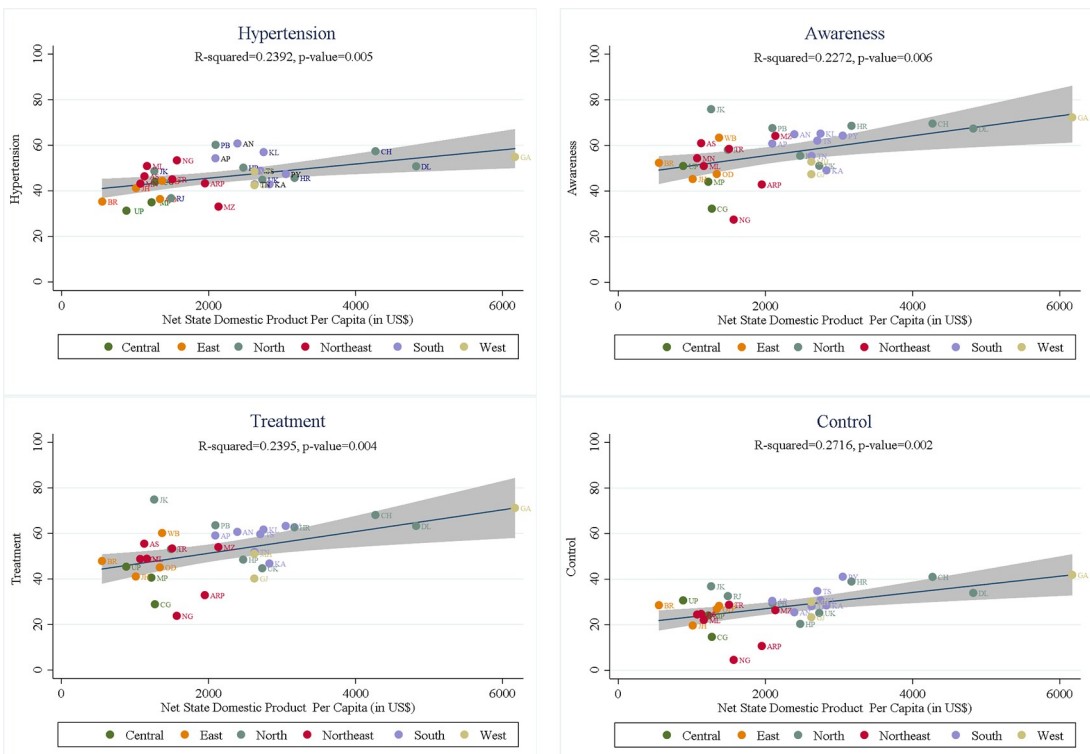

**Fig 3. The association of NSDP per capita (in US$) with each of hypertension prevalence and ATC.** Note: NSDP per capita for the year survey period (2017–2018) were taken from the Reserve Bank of India website and converted into US$ (1 US$ = ₹66.75) (https://rbi.org.in/Scripts/AnnualPublications.aspx%3Fhead%3DHandbook%20of%20Statistics%20on%20Indian%20States). AN, Andaman and Nicobar Islands; AP, Andhra Pradesh; ARP, Arunachal Pradesh; AS, Assam; ATC, awareness, treatment, and control; BR, Bihar; CG, Chhattisgarh; CH, Chandigarh; DL, Delhi; GA, Goa; GJ, Gujarat; HR, Haryana; HP, Himachal Pradesh; JH, Jharkhand; JK, Jammu and Kashmir; KA, Karnataka; KL, Kerala; MH, Maharashtra; ML, Meghalaya; MN, Manipur; MP, Madhya Pradesh; MZ, Mizoram; NG, Nagaland; NDSP, net state domestic product; OD, Odisha (Orissa); PB, Punjab; PY, Puducherry; RJ, Rajasthan; TN, Tamil Nadu; TR, Tripura; TS, Telangana State; UK, Uttarakhand (Uttaranchal); UP, Uttar Pradesh; WB, West Bengal.

after controlling for other characteristics and state. Adding controls did not eliminate socioeconomic inequalities in ATC. For example, the difference between the richest fifth and the poorest fifth was estimated to be 8.5 pp (95% CI 5.3 to 11.7), 8.9 pp (95% CI 5.7 to 12.0), and 7.1 pp (95% CI 4.2 to 10.1) for ATC, respectively. These outcomes all remained higher among females, urban dwellers, and the nonemployed after conditioning on other characteristics. Awareness and treatment, but not control, were higher among older adults holding other characteristics fixed.

## Discussion

In India, hypertension is the main risk factor for CVD [10], accounts for the largest disease burden of any condition [3], and is substantially more prevalent among older adults [8]. Yet, to our knowledge, there were no nationally representative and state-specific estimates of hypertension ATC for the older Indian population. This study estimated high prevalence of hypertension and found substantial gaps in its diagnosis, treatment, and control in the older population of India in 2017 to 2018. We estimated that 42% of adults aged 45 years and over and their spouses had hypertension, while only half of those with the condition had been diagnosed and treated, and less than one-third had achieved BP control.

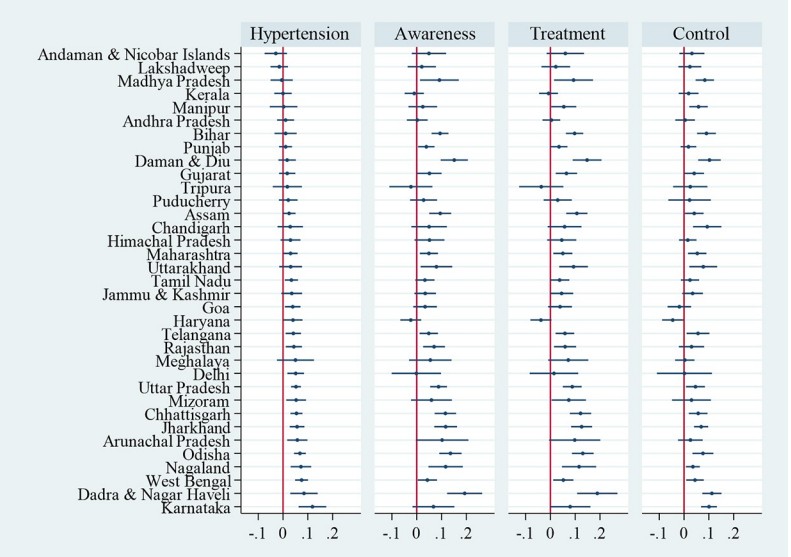

**Fig 4. Adjusted concentration indices for hypertension and ATC among those with hypertension by states, adults aged 45 years and over and their spouses.** Notes: A concentration index is a scaled covariance between an outcome and rank in the distribution of MPCE. Adjusted for age and sex. States ordered by concentration indices for hypertension. Dots indicate point estimates, and horizontal lines show 95% CIs. State-specific sample sizes presented in S5 Table and estimates in table format presented in S11 Table. ATC, awareness, treatment, and control; MPCE, monthly per capita consumption expenditure.

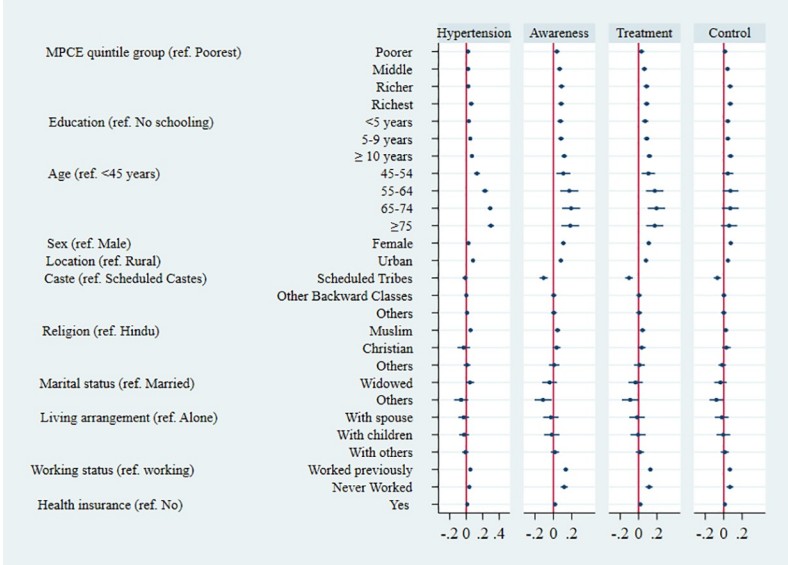

**Fig 5. Averaged marginal effects on probability of hypertension and on probabilities of ATC among those with hypertension, adults aged 45 years and over and their spouses.** Notes: Estimated averaged marginal effects on probability of the respective outcome from multivariable logistic regressions. Dots show point estimates. Horizontal lines show 95% CI. Numerical estimates are presented in table format in S12 Table. All regressions also include a complete set of state indicators (fixed effects) in addition to all the covariates listed in the table. Estimates of state-specific averaged marginal effects presented in S12 Table. ATC, awareness, treatment, and control; MPCE, monthly per capita consumption expenditure.

We found substantial geographic and socioeconomic inequalities in the prevalence of hypertension and in its diagnosis, treatment, and control. The states with the highest prevalence included those most advanced in the demographic transition, such as Kerala, and the high-income states, such as Goa and the capital Delhi. While prevalence tended to be lower in the low- or middle-income states and it was relatively low in all 4 states with the highest rates of poverty (Bihar, Jharkhand, Madhya Pradesh, and Uttar Pradesh) [21]—these states also achieved lower rates of ATC. Consequently, we would expect cross-state variation in CVDs arising from uncontrolled hypertension to have been less strongly associated with economic development than was hypertension itself. Those living in poorer states were at lower risk of hypertension. If they were hypertensive, however, they were also more vulnerable to diseases associated with the risk factor because they were less likely to have been diagnosed, treated, and controlled. However, the examples of Jammu and Kashmir and West Bengal demonstrated that higher rates of ATC could be achieved even if lower incomes constrained available resources. The reasons why some states performed better than others at similar levels of economic development deserve further attention. Previous estimates of ATC rates in the older Indian population that used data on these outcomes for older people in only 6 states [17] relied on the strong assumption that the patterns observed in those states would hold in other states. We showed substantial variation across states in rates of ATC and in the socioeconomic gradient of each outcome.

The sociodemographic patterns among individuals were consistent with the patterns observed across states. While socioeconomically disadvantaged individuals were less likely to have hypertension, if they did have it, they were less likely to have been diagnosed and treated and to have achieved BP control. Among those with hypertension, only 42% of those in the poorest fifth of the sample had been diagnosed, while 62% of the richest fifth were diagnosed. Consequently, the socioeconomic gradient observed in the prevalence of hypertension partly reflected the lower propensity of poorer people to be diagnosed. This pattern is consistent with the finding that states with greater socioeconomic inequality in awareness also tended to have greater inequality in prevalence.

Among those with hypertension, almost 60% of those in the richest fifth were receiving treatment, while only 40% of those in poorest fifth were treated. Differences in the propensity to be diagnosed and treated imply that the higher prevalence of hypertension observed in richer and better educated groups overstated socioeconomic inequality in exposure to the risk of CVDs that is associated with unmanaged hypertension.

Rates of hypertension ATC were lower not only for poorer and less educated study participants but also for those who were relatively younger (not control), male, rural dwellers, widowed or unmarried, and working. These characteristics define the profile of older Indian adults who appear to have least access to diagnosis, medication, and knowledge of how to control hypertension. The same characteristics identify those with the lowest estimated hypertension prevalence, which further suggests that apparent differences in prevalence partly reflect differences in the likelihood of being diagnosed and treated and that patterns of prevalence should not exclusively determine the targeting of hypertension management interventions.

Our estimates of hypertension prevalence in India among adults aged 45 years and over and their spouses are slightly higher than estimates based only on BP measurement obtained from nationally representative data for comparable age groups (S13 Table) [8]. The differences likely stem from our identification of hypertension using reported diagnosis and medication/diet restriction to control BP in addition to measured BP (S13 Table). The higher prevalence we found among older adults located in high-income states and belonging to socioeconomically advantaged groups are consistent with the patterns previously observed in a very large sample of Indian adults aged 18+ [8]. That study could not estimate hypertension ATC, and

the only previous nationally representative study of these outcomes was restricted to the 15 to 49 age range [13]. Compared with that study, we estimated a higher rate of hypertension awareness, and we estimated rates of treatment and control that are substantially higher. The differences are partly explained by the facts that we examined a much older age range and found, like other studies [17,18], that awareness and treatment, but not control, increased with age. Like many small, local studies of ATC in India [14–17], we found higher rates in urban areas than in rural areas. We estimated considerably higher rates of ATC than were obtained from a meta-analysis of small-scale Indian studies up to 2013 [15]. The differences may be partly attributable to the fact that we examined an older age group, as well as any improvements in the management of hypertension that have arisen from the NCD initiatives taken by the Indian government since 2013 [12]. Our findings of lower rates of ATC among men and socioeconomically disadvantaged groups are consistent with other Indian studies [18,19].

In 2013, the Government of India adopted a national action plan for prevention and control of NCDs and set an ambitious target to reduce hypertension prevalence [12]. However, the India Hypertension Control Initiative (IHCI) was launched—in 5 states—only in November 2017. An observational study found that BP control achieved by public health facility patients increased from 26% at registration in the IHCI to 60% at follow-up [26]. Our findings can help identify states and sociodemographic groups to be targeted as this program is rolled out over the country. That targeting should balance attention to prevalence, which is higher in the high-income states and socioeconomically advantaged groups, with attention to gaps in ATC, which are greater in the low- or middle-income states and disadvantaged groups.

Low awareness of hypertension in the older population of India signals a pressing need for improved health education and screening. The national NCD action plan [12] stipulates opportunistic screening for hypertension at public health facilities, which needs to be implemented effectively and extended to the more frequently used private clinics. Low rates of treatment and control of hypertension, particularly among poorer individuals and in rural areas, suggest difficulties in accessing and affording primary healthcare. Although low cost generic antihypertensives are widely available, they may not be affordable for poorest people. Non-pharmacological interventions, on the other hand, may be hard to adhere to in the long run. The large interstate variation we found in hypertension ATC presumably arises, in part, from the fact that health is a state responsibility in India. While the central government can issue guidelines and provide financial assistance for certain health services, state governments are largely responsible for the financing and management of facilities, manpower, and drug supply. Greater investment by both national and state governments in hypertension management is infeasible in the short term as the health system is straining to cope with the Coronavirus Disease 2019 (COVID-19) epidemic. But the fact that older adults with chronic conditions, such as hypertension, are most likely to succumb to this illness and need high-cost medical care emphasizes that investment in hypertension screening and management can potentially pay off in the long term through reduced demands on the health system.

The main limitation of this study, which is common to all ATC care cascade studies [14,16,18] and most prevalence studies [8,27,28], is that hypertension status was identified from BP measured on a single occasion. Some of the participants with a high BP reading will have been false positives. This is likely to have pushed the estimates toward overestimation of hypertension prevalence and underestimation of rates of awareness and treatment among those who had hypertension. We could think of no reason for this measurement error to have varied systematically across sociodemographic groups or states, and so it may not have biased estimates of socioeconomic and geographic inequalities in awareness and treatment. BP could not be measured for almost 9% of LASI participants. While this percentage is not especially high for a general purpose population survey of older adults, the fact that BP measurements

were slightly less likely to be obtained from some sociodemographic groups (S1 Table) that had higher hypertension prevalence and rates of ATC may have pushed the estimates of population averages toward some underestimation of both prevalence and ATC. Taken together, these two data limitations arising from measurement of BP on a single occasion and systematic differences in non-measurement of BP are likely to have had offsetting effects on the estimate of hypertension prevalence but potentially compounding, downward effects on estimates of ATC. We may have underestimated the extent to which hypertension is diagnosed, treated, and controlled in the older Indian population, on average. However, this potential bias in the estimates of the averages does not call into question the findings that rates of ATC are substantially lower in less privileged socioeconomic groups and differ substantially across states.

A third limitation is that the study did not collect data on medication adherence, and so it is not possible to examine the extent to which this contributed to the low rate of controlled BP. Participants were not asked if they had ever had their BP measured. This prevented us from both examining screening as first step in the hypertension care cascade and distinguishing between the unaware who had never been screened and those unaware despite having had their BP measured. While the main contribution of this study was to add nationally representative evidence on hypertension ATC in the older population of India in which the condition is more prevalent, a limitation is that the sample did not include adults aged below 45, except for spouses. Given India's large and young population, there is a large absolute number of younger adults with hypertension in the country [8]. Our analysis did not extend to this population, although the age differences we observed would suggest that awareness and treatment, and, possibly, also BP control, may well have been even lower at younger ages. Prenissl and colleagues [13] provided evidence on hypertension ATC at aged 15 to 49, and this study should be viewed as complementary to that one.

A final limitation, which is also common to other care cascade studies, is the presumption of a "treat-to-target" approach to hypertension management with the target set to controlling BP <140/90 mm Hg. Recent guidelines have defined hypertension at lower thresholds [29], aimed for BP control at lower levels [29,30], and advocated a "benefit-based" approach that involves offering hypertension therapy to patients at high CVD risk even if their BP is below the hypertension threshold [29,31,32]. This new approach is based on evidence that even someone with "normal" BP according to the thresholds used in this study could benefit from BP therapy that substantially reduces their elevated CVD risk. It suggests that the evidence we presented may understate gaps that existed in the treatment and control of BP in India [33].

Notwithstanding these limitations, to our knowledge, this study produced the first estimates of hypertension ATC using nationally representative data on older people spread throughout the whole of India. It described socioeconomic and geographic inequalities in ATC that can be used to motivate policy action on hypertension management, target interventions, and monitor their effects.

## Supporting information

**S1 Checklist. STROBE checklist of items that should be included in reports of cross-sectional studies.**
(DOC)

**S1 Text. Sample design.**
(DOCX)

**S2 Text. Outcome definitions.**
(DOCX)

**S3 Text. Measurement of MPCE. MPCE, monthly per capita consumption expenditure.**
(DOCX)

**S4 Text. Age–sex adjustment.**
(DOCX)

**S1 Fig. Adjusted concentration curves for hypertension cases and ATC among those with hypertension, adults aged 45+ and their spouses in India.** The figure shows concentration curves, which depict relative inequality in an outcome in relation to a measure of socioeconomic status (O'Donnell and colleagues 2008). For example, the curve in the top-left panel traces the cumulative proportion of hypertension cases (y-axis) against the cumulative proportion of participants ranked from the poorest (left) to the richest (right) (x-axis) based on MPCE. The other curves trace the cumulative proportion of participants with hypertension who are aware, treated, and controlled (y-axis) against the cumulative proportion ranked from poorest to richest. Shading around the curves indicates 95% CIs. Each curve lies below the respective 45-degree line, which indicates that there is a disproportionate concentration of hypertension cases among richer participants and that among those with hypertension, ATC are also disproportionately concentrated among the richer participants. ATC, awareness, treatment, and control; MPCE, monthly per capita consumption expenditure.
(TIF)

**S2 Fig. Adjusted percent treated among those aware of hypertension and adjusted percent controlled among those treated, adults aged 45+ and their spouses in India.** The base map can be found at https://globalsolaratlas.info/download/india. Adjusted for age and sex.
(TIF)

**S1 Table. Sample characteristics by whether the BP was measured, adults aged 45+ and their spouse, adults aged 45+ and their spouses.**
(DOCX)

**S2 Table. Missing observations on BP measurement, diagnosis, treatment and sociodemographic variables, adults aged 45+ and their spouses.**
(DOCX)

**S3 Table. Estimates of hypertension prevalence and rates of ATC from full item response analysis sample and alternative sample including participants missing on MPCE or any sociodemographic variable, adults aged 45+ and their spouses.**
(DOCX)

**S4 Table. Number of participants with hypertension by sociodemographic characteristics.**
(DOCX)

**S5 Table. Number of participants in full analysis sample and with hypertension by state.**
(DOCX)

**S6 Table. Unadjusted estimates of hypertension prevalence and percent aware, treated, and controlled among those with hypertension by sociodemographic characteristics, adults aged 45+ and their spouses.**
(DOCX)

**S7 Table. Adjusted prevalence of undiagnosed, untreated, and uncontrolled hypertension by MPCE quintile and education groups, adults aged 45+ and their spouses in India.**
(DOCX)

**S8 Table. Adjusted hypertension prevalence, and percent aware, treated, and controlled among those with hypertension by state, adults aged 45+ and their spouses in India.**
(DOCX)

**S9 Table. Unadjusted estimates of hypertension prevalence and percent aware, treated, and controlled among those with hypertension by state, adults aged 45+ and their spouses in India.**
(DOCX)

**S10 Table. Correlation of rates of ATC each with hypertension prevalence across states, adults aged 45+ and their spouses.**
(DOCX)

**S11 Table. Adjusted concentration indices for hypertension and ATC among those with hypertension, adults aged 45+ and their spouses in India.**
(DOCX)

**S12 Table. Adjusted marginal effect for hypertension and ATC among those with hypertension, adults aged 45+ and their spouses in India.**
(DOCX)

**S13 Table. Comparison of estimates of hypertension prevalence by age and sex with those in Geldsetzer and colleagues.**
(DOCX)

**S14 Table. Numbers of participants by age and sex used to estimate prevalence rates presented in S13 Table.**
(DOCX)

**S1 Code. Stata code for computation of prevalence and ATC of hypertension.**
(TXT)

## Author Contributions

**Conceptualization:** Sanjay K. Mohanty, Jürgen Maurer, Owen O'Donnell.

**Data curation:** Ashish Kumar Upadhyay, Prashant Shekhar, Radhe Shyam Mishra.

**Formal analysis:** Sanjay K. Mohanty, Ashish Kumar Upadhyay, Prashant Shekhar, Radhe Shyam Mishra, Owen O'Donnell.

**Funding acquisition:** Fabrice Kämpfen, Jürgen Maurer.

**Investigation:** Sanjay K. Mohanty, Ashish Kumar Upadhyay.

**Methodology:** Owen O'Donnell.

**Project administration:** Sanjay K. Mohanty, Sarang P. Pedgaonkar, Fabrice Kämpfen, Jürgen Maurer, Owen O'Donnell.

**Resources:** Fabrice Kämpfen, Radhe Shyam Mishra.

**Software:** Ashish Kumar Upadhyay, Owen O'Donnell.

**Supervision:** Jürgen Maurer.

**Validation:** Owen O'Donnell.

**Visualization:** Sarang P. Pedgaonkar, Prashant Shekhar.

Writing – **original draft:** Sanjay K. Mohanty, Owen O'Donnell.

Writing – **review & editing:** Sarang P. Pedgaonkar, Fabrice Kämpfen, Jürgen Maurer, Owen O'Donnell.

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
