## [Editor Report · Decision Letter 0]

20 May 2021

Dear Dr Mohanty, 

Thank you for submitting your manuscript entitled "Awareness, treatment, and control of hypertension in India: a nationally representative cross-sectional study of older adults aged 45+" for consideration by PLOS Medicine.

Your manuscript has now been evaluated by the PLOS Medicine editorial staff as well as by an academic editor with relevant expertise and I am writing to let you know that we would like to send your submission out for external peer review.

Please re-submit your manuscript within two working days, i.e. by May 24 2021 11:59PM.

Kind regards,

Callam Davidson

Associate Editor

PLOS Medicine

---

## [Decision Letter · Decision Letter 1]

16 Jun 2021

Dear Dr. Mohanty,

Thank you very much for submitting your manuscript "Awareness, treatment, and control of hypertension in India: a nationally representative cross-sectional study of older adults aged 45+" (PMEDICINE-D-21-02188R1) for consideration at PLOS Medicine. 

Your paper was evaluated by an associate editor and discussed among all the editors here. It was also discussed with an academic editor with relevant expertise, and sent to independent reviewers, including a statistical reviewer. The reviews are appended at the bottom of this email and any accompanying reviewer attachments can be seen via the link below:

[LINK]

In light of these reviews, we will not be able to accept the manuscript for publication in the journal in its current form, but we would like to invite you to submit a revised version that addresses the reviewers' and editors' comments fully. You will appreciate that we cannot make a decision about publication until we have seen the revised manuscript and your response, and we expect to seek re-review by one or more of the reviewers. 

We hope to receive your revised manuscript by Jul 07 2021 11:59PM. Please email us (plosmedicine@plos.org) if you have any questions or concerns.

Please let me know if you have any questions, and we look forward to receiving your revised manuscript. 

Sincerely,

Callam Davidson, 

Associate Editor, 

PLOS Medicine

plosmedicine.org

Please can the authors clarify their response ('No - some restrictions will apply) to the Data Availability question in the introductory form, as it appears that the data used are publicly available.

Please add a completed STROBE/RECORD checklist as a supplementary file, labelled "S1_STROBE_Checklist" or similar and referred to as such in the Methods section. 

In the checklist, please refer to individual items by section (e.g., "Methods") and paragraph number, not by line or page numbers as these generally change in the event of publication. 

Please update the manuscript to include line numbering in the margins.

Please update the manuscript title such that the age range studied comes before the colon (e.g. Awareness, treatment, and control of hypertension in adults aged 45+ in India: a nationally representative cross-sectional study)

Please add a new final sentence to the "Methods and findings" subsection of your abstract, which should begin "Study limitations include..." or similar and should quote 2-3 of the study's main limitations. 

Please add basic sample demographic information to the abstract (median age and % male/female at a minimum).

Please present conclusions in the past tense (e.g. in "Conclusion" section of abstract - 'Hypertension prevalence was high' rather than 'is high'. Please check throughout manuscript.

Please change the style of all in-text citations. These should be in square brackets and occur before punctuation. 

Please remove the information on funding and data availability from the main text. In the event of publication, this information will appear in the article metadata via entries in the submission form. 

Please remove the section 'Role of the funding source' from the manuscript.

Please add 1-2 sentences at the beginning of the discussion summarising why the study was performed followed by the major findings.

The discussion sentence introduces new data that is not described in the results section text (notably Spearman rank correlation rho and associated p values). Please ensure that any results discussed in the discussion section are first described adequately in the results section. 

In the final paragraph of the discussion, please update the first sentence to include '...to our knowledge, this study produced the first estimates of hypertension ATC...' 

Please ensure references do not contain bold or italicised text, and please note that references should list the first six authors followed by et al. where appropriate.

If an analysis plan was prepared prior to the study, please could this be provided in the supplementary materials. 

Please confirm that the maps used to produce Figure 3 and supplementary figure D2 are appropriate for publication under a Creative Commons CC-BY licence.

In tables that state P values, please update any P = 0.000 values to P < 0.001. 

Comments from the reviewers:

Reviewer #1: This is a cross-sectional analysis of hypertension "care cascades" using nationally representative data from older adults in India collected in the most recent LASI cohort wave. The study is well conducted, advances knowledge on the epidemiology of hypertension care in India, and is well-written in its framing relative to prior literature. I think the paper would be a good fit at PLOS Medicine but also could be improved through several suggested revisions.

1. The authors should provide clinical justification for why they have selected a BP threshold of <140/90 to define hypertension and hypertension control. I am not familiar with hypertension management guidelines in India, but this is a controversial topic and major guidelines including the ACC/AHA and WHO-PEN now recommend more strict thresholds in individuals with higher cardiovascular risk. Please see my comment below on limitations for more on my perspective on this issue.

2. I agree with primarily presenting sex- and age-adjusted estimates in the state-level data. However, the methods for age-sex adjustments for hypertension prevalence, awareness, treatment, and control is not 100% clear to me. Am I correctly inferring that the authors ran the logistic regression models with state fixed effects as well as fixed effects for sex + age groups (categorical rather than continuous age), and then used "average adjusted predictions" for each state?

3. The authors mention that they are enclosing their Stata .do files. This would be useful and welcome. I do not see the files in the submission, however, so I have been unable to personally review any statistical code.

4. Figure 3: would strongly urge the others to redraw the heatmap using color gradients rather than color tertiles for proportions by state. Much granularity of information is lost using the tertiles.

5. This is a stylistic suggestion, but I would recommend moving the table of odds ratios (Table 4) to the appendix and instead present a table of average marginal effects from the logistic models. Such a table would be much easier for readers to interpret the absolute differences in proportions within categories of covariates. Additionally, in place of a table, I would favor using forestplots to present regression outputs visually; in Stata, "coefplot" is a very nice package to do this, if the authors would like to pursue this route.

6. I would request that the authors add a supplementary appendix specifying the precise survey questions that were used in LASI to define awareness and treatment. This can be a simple table and will help readers understand the numerators for the cascades as you define them.

7. Some states had higher proportions of individuals controlled than treated. I believe that this is because the authors have not defined the hypertension cascades to be conditional, i.e., achieving one step was not necessarily contingent on the prior step. For example, an individual could have reported a prior diagnosis of hypertension, not be receiving treatment, and yet still have a BP level under the controlled threshold. Such unconditional cascades are reasonable (and my own personal preference in these analyses), though I believe the Prenissl India study used conditional cascades. Since the finding of % control > % treated appears a bit unanticipated, especially to a reader less familiar with the definitional nuances of these studies, the authors may wish to add 1-2 lines in either a caption or main text describing why this finding can occur.

8. The authors have defined hypertension treatment strictly as pharmacological treatment. Why? Non-pharmacological therapy is effective for lowering blood pressure in clinical trials, and, LASI also had at least one question specifically on lifestyle changes for treatment of blood pressure (diet/salt focused). I believe the authors should either (1) specifically state why they opted to exclude non-pharmacological treatment in their estimates, (2) include diet as "treatment" in the main analysis, or (3) generate a sensitivity analysis in which diet treatment was also considered "treatment."

9. I wonder if a missing opportunity in this manuscript is at least exploring some state-level characteristics that are associated with hypertension prevalence and ATC. This need not be a complex formal analysis. For example, the authors could consider simply plotting proportions by state-level GDP or some other marker of economic development (as in the Prenissl paper). The authors comment on this already in the discussion (second paragraph, bottom of page 10).

10. The supplementary tables E11 and E12 are very useful in helping readers locate this study in the context of the prior Geldsetzer paper in JAMA IM. Thanks for including it.

11. I can think of a few additional limitations that the authors may wish to mention:

* I agree that measurement error is probably the biggest limitation of this and other cross-sectional assessments of hypertension. I might just also add text that the hypertension prevalence is likely overestimated. This idea is suggested but not specifically stated.

* Lower age limit of LASI <45 years cuts off a substantial absolute number of adults with hypertension at younger ages, given that India has a relatively young population age distribution.

* LASI does not assess if a respondent has ever had blood pressure measured (sometimes worded as "ever screened" in cascade papers). This is often a step in the hypertension cascade that is reported when available.

* I mentioned the issue of non-pharmacological therapy above. If the authors do not wish to address this in the analysis, then it should be mentioned as a limitation.

* A conceptual limitation of the present study is that it focuses on the "treat-to-target" approach for hypertension treatment. As the authors may be aware, the field has been moving away from this approach to hypertension treatment, which is the one operationalized in this study. This shift has happened due to evidence that the relative benefits of hypertension therapy are constant across risk levels; thus, individuals with higher baseline risk have much larger absolute benefit irrespective of blood pressure. In other words, even a person with "normal" blood pressure according to traditional BP thresholds may have large benefit from BP therapy. This approach is sometimes referred to as "benefit-based" therapy and has begun to be incorporated into recent guidelines from the ACC/AHA and WHO-PEN.

Reviewer #2: I confine my remarks to statistical aspects of this paper. The general approach is fine, but I do have some questions and issues to resolve before I can recommend publication.

NOTE: Line numbers would have made the review process easier.

p. 4 In addition to "44% of global deaths ...." give a number of deaths, or rate per 100,000 or something.

p. 5 While your results should definitely be useful, it IS possible to estimate these numbers using the analyses that have already been done. Those estimates might be poor, but, for instance, you could estimate the rates for older people in districts where they are not known by adjusting the rates in known districts using the more general results that were gotten in all districts.

p. 6 It would be good to clarify that you can be a, b, and c .... or other combinations. It seems to me that you can be a) alone, a) and b), or a), b) and c). But it would aid clarity to spell this out.

p. 7 Why did you use the concentration index? What does it add to the regression results? It's a pretty unusual statistic, so there should be some reason for using it. 

 Why was BP unmeasurable on some people? Were these people different from those where it could be measured? Why wasn't multiple imputation used? (I think it should be, but you may have some reason for not doing so)

Figure 3: It would be better to use more colors and to have the percent relate to some quality of the color (e.g. saturation). A very good site on doing this well is ColorBrewer https://colorbrewer2.org/#type=sequential&scheme=BuGn&n=3

Peter Flom

Reviewer #3: The authors provide a clear picture of hypertension prevalence and rates of awareness, treatment, and control among older adults in India. The research idea is a valuable extension of existing evidence on younger age groups and highlights the need for better hypertension care among the population segment most at risk of suffering from hypertension and cardiovascular diseases. 

However, the definition of their sample needs to be improved. The authors defined participants who have "ever […] been diagnosed as hypertensive" (p. 6) to have hypertension. The survey instrument does not distinguish between hypertension and high blood pressure. Thus, this is a rather broad indicator, which does not reflect clinical guidelines and can capture temporary episodes of high blood pressure. I assume that the inclusion of "having been diagnosed" in the study sample is the reason why in some age groups and states the control rates are larger than the treatment rates (Table E3, Bihar, for example). The authors should base the definition of hypertension on BP and self-reported medication use only.

Additional specific comments:

- Last sentence in first paragraph of introduction (p.4) is too long.

- Average of BP measures (p.6.): were there missing values in either the second or the third measurement? If yes, it needs to be stated how the final BP was computed if only one of the two measurements was available. 

- MPCE indicator (p. 6, "Measures"): It is not clear to me why all expenditure on healthcare and medicine were excluded from the calculation. This should briefly be explained. 

- Levels of Covariates (p.6, "Statistical Analysis"): The main text, appendix or notes of Table 1 should include an overview on how levels of categories were chosen and which information they include. Eg. what categories were clubbed in the "other" groups, how were years of education cut-offs defined. Furthermore, it should be considered if the categories in "other" really are comparable or should better be listed separately, despite their small sample size. 

- Missingness in variables of interest (p. 7, "Results"): Roughly 10% of observations have either a missing value in the BP measurement or another variable. The appendix should include a comparison of the sample characteristics of the samples with and without missings in the variables of interest (BP measurement, ATC variables). A detailed table of the number(%) of missing values in each outcome and covariate should be provided. 

- Median age (p.8): IQR should be reported alongside the median. 

- Thresholds for comparison (p.9 "Figure 2 shows […] in Haryana"): The threshold for comparison, eg "the rate of awareness exceeded 70% in only three states…" seem arbitrary and are confusing. I suggest deleting these kinds of comparisons.

- Figure 3: The thresholds for the colour coding vary in each panel. Either, the same cut-offs or different colours in each panel should be used. Or it should be made clearer that the focus is on the relative comparison across states, e.g. by removing the prevalence ranges from the legend. The same applies to the other maps of this kind.

- Multivariate regression (p. 10): I assume the authors mean "multivariable logistic regression"

- Presentation of results in Discussion section: The authors present new results in the discussion. The discussion should only briefly summarize the main results (without presenting p-values and CIs). The focus should be on the interpretation of the results and putting them into context. 

- Comparison to previous evidence from other studies in Discussion: This is interesting but too detailed. A higher level comparison would be more suitable. 

- In several instances, the authors refer to the poverty rates across states. It would be interesting to include this information in figures/tables; eg. by colour coding Figure 4 according to poverty rate quartile. 

- Decimal places: The authors should report only two decimal places, also when presenting the concentration index results.

- Oxford comma: The Oxford comma should be used consistently. 

Reviewer #4: 1. The authors used some acronyms in the abstract, which can be avoided if not well-known ones like ATC.

2. About 10% of participants were excluded from the study due to some reasons. There may be a possibility that the excluded participants may have different socioeconomic characteristics. Authors may check that the participants included in the study are not different from excluded participants. 

3. The initial recommendation of hypertension control is lifestyle modification. If this information is available, authors may think about modifying treatment definition in the cascade of care.

4. "STATA" is a typo. Please have a look: https://www.stata.com/statalist/archive/2010-01/msg00489.html

5. Authors mentioned, "Table 1 shows characteristics of the full analysis sample and estimates of age-sex adjusted hypertension prevalence. The median age of the sample was 57 years." The measure of central tendency gives only a half picture. Please use a measure of dispersion along with a measure of central tendency here and throughout the manuscript. 

6. The study's title, all the tables, and figure captions indicate that analysis is only for 45+ aged participants, but about 9% of participants were <45 years as per table 1. Please make it clear. 

7. The authors used Spearman rank correlation to measure the relationship between hypertension prevalence, awareness, and treatment. Some other methods may be preferred in such cases.

8. Figure 2 and figure 3 seems duplication of information; authors may choose one of them. 

9. In Figure 3, the authors categorized the prevalence of hypertension prevalence and cascade of care, masking the state-level variation. It may be worth to show it using a continuous scale. 

10. Authors mentioned, "Table 4 shows estimated OR from a multivariate logistic regression for each outcome." Although it is used as a synonym, there is a difference in multivariable and multivariate regression. Authors may recheck that they used multivariate or multivariable regression.

11. In table 4, the authors categorized age in various age groups, which can lead to loss of information and an increase in type I and II errors. A supplementary analysis is warranted in the case of categorization of continuous data. 

12. The authors included caste and religion as independent variables in table 4, but biologically caste and religion have nothing to do with hypertension and its cascade of care. Authors may think about removing caste and religion or may be interested in assessing the impact of all other variables excluding caste and religion in a separate analysis. 

13. Authors mentioned, "The discrepancies are partly explained by the facts that we examined a much older age range and found, like other studies." Discrepancy may not be the correct word choice as the differences are obvious with higher age, especially in postmenopausal women.

[LINK]

---

## [Decision Letter · Decision Letter 2]

19 Jul 2021

Dear Dr. Mohanty,

Thank you very much for re-submitting your manuscript "Awareness, treatment, and control of hypertension in adults aged 45+ and their spouses in India: a nationally representative cross-sectional study" (PMEDICINE-D-21-02188R2) for review by PLOS Medicine.

I have discussed the paper with my colleagues and the academic editor and it was also seen again by three reviewers. I am pleased to say that, provided the remaining editorial and production issues are dealt with, we expect to be able to accept the paper for publication in the journal.

[LINK]

Please let me know if you have any questions, and we look forward to receiving the revised manuscript. 

Sincerely,

Callam Davidson, 

Associate Editor 

PLOS Medicine

cdavidson@plos.org

Requests from Editors:

Please update ‘aged 45+’ to ‘aged 45 years and over’ throughout the manuscript (including in the title).

Thank you for responding to my previous query regarding your data availability statement, it appears that your data sharing approach aligns with the journal policy. While you have clarified the matter for me, please can you update your response to read ‘No – some restrictions will apply’ as it did in the first revision, as this is more accurate. 

In the ‘Describe where data can be found’ section of the submission form:

* you state ‘Data is publicly available’ – please update to ‘Data are publicly available’. Please update elsewhere in the manuscript if necessary to ensure data are treated as plural.

* Please add a sentence to the effect of ‘data will be made available to the researchers meeting the criteria for access to confidential data’, if applicable and accurate.

In both the abstract and discussion, please update the sentence ‘Study limitations included reliance in BP’ to ‘Study limitations included reliance on BP’.

Please provide 95% CIs and p values for quantitative results presented in the abstract, as appropriate.

Please consider whether any quantitative data ought to be added to the abstract to support the results detailed on lines 29-32. If you opt to include, please ensure it is presented with 95% CIs and p-values, where appropriate.

The first sentence of the ‘Notes’ under Table 1 (‘Adjusted for age and sex, except age (sex) group estimates adjusted for sex (age) composition’) is confusing, please reword to improve clarity. 

In several tables and figures, you need to define the abbreviation MPCE in the legend. Please check and update throughout.

Table 3 still contains p-values of 0.000. Please update these to P<0.001.

Please ensure consistent use of ‘Figure’ (as opposed to ‘Fig’).

On lines 328-329, please report the actual p-values rather than stating p-value<0.05. 

The first line of your discussion should be split into two sentences. I would advise placing a full stop after [8] on line 362, then beginning the following sentence ‘Yet, to our knowledge, there were no nationally representative…’ etc.

Please add the date accessed for references 24 and 31.

Please review your references and ensure that initials are not followed by a full stop (as in, for example, reference 32), and also please ensure journal titles are abbreviated consistently (e.g. reference 33 should be ‘Circ Res’.

Comments from Reviewers:

Reviewer #1: I have reviewed the author responses in detail. My concerns have been very thoughtfully addressed, and I have no further critiques.

I wish to commend the authors for making their full statistical code available and for generally completing a very thoughtful revision.

This is a very nicely done paper and the authors should be proud.

Reviewer #2: The authors have addressed my concerns and I now recommend publication. 

Reviewer #4: As authors have addressed almost all of my comments, I recommend the acceptance of the paper.

[LINK]

---

## [Editor Report · Decision Letter 3]

23 Jul 2021

Dear Dr Mohanty, 

On behalf of my colleagues and the Academic Editor, Dr Sanjay Basu, I am pleased to inform you that we have agreed to publish your manuscript "Awareness, treatment, and control of hypertension in adults aged 45 years and over and their spouses in India: a nationally representative cross-sectional study" (PMEDICINE-D-21-02188R3) in PLOS Medicine.

Please also make the following changes to your manuscript when completing your formatting changes:

* Please delete the heading 'Title Page' from the top of your title page 

* Please remove the words 'the first' from line 90 in your Author Summary

* Please remove the words 'and growing' from line 110 in your Author Summary

* Please refer to low or middle income states rather than "developing states". Please refer to high income states rather than "developed" states (update as appropriate on lines 115, 117, 336, 342, 411, 412, 456, 477, 479).

* Please update Data Availability Statement to read 'Data can be obtained from https://iipsindia.ac.in/ by registering at https://iipsindia.ac.in/sites/default/files/LASI_DataRequestForm_0.pdf. Data will be made available to the researchers meeting the criteria for access to confidential data. Questions can be directed to datacenter@iipsindia.ac.in'

* Please note that the 'DO' file you refer to in your Data Availability Statement is now a TXT file. Either update the wording to 'we are enclosing the DO file information as a TXT file' or provide as a DO file per the original submission. Please also specify exactly which graph you are referring to in your data availability statement (such that the reader could locate it). 

PRESS

Sincerely, 

Callam Davidson 

Associate Editor 

PLOS Medicine

cdavidson@plos.org